# Excessive Dental Bleaching with 22% Carbamide Peroxide Combined with Erosive and Abrasive Challenges: New Insights into the Morphology and Surface Properties of Enamel

**DOI:** 10.3390/ma15217496

**Published:** 2022-10-26

**Authors:** Edson de Sousa Barros Júnior, Mara Eliane Soares Ribeiro, Rafael Rodrigues Lima, Mário Honorato da Silva e Souza Júnior, Sandro Cordeiro Loretto

**Affiliations:** 1Biomaterials Laboratory of the Postgraduate Program in Dentistry, Institute of Health Sciences, UFPA, Belém 66075-110, Pará, Brazil; 2Laboratory of Functional and Structural Biology, Institute of Biological Sciences, UFPA, Belém 66075-110, Pará, Brazil

**Keywords:** enamel, tooth bleaching, dental erosion, dental abrasion

## Abstract

This study aimed to evaluate the effects of 22% carbamide peroxide combined with an erosive challenge and simulated brushing on enamel. Bovine incisor teeth were divided into G1, tooth bleaching; G2, tooth bleaching + erosive challenge; and G3, tooth bleaching + erosive challenge + simulated brushing, and evaluated at T0, before any intervention; T1, 14 days after the proposed treatments; and T2, 28 days after the proposed treatments. Tooth bleaching was performed daily for 1 h for 28 days. The microhardness, surface roughness, mass variation, and ultrastructure were analyzed at T0, T1, and T2. Two-way analysis of variance for repeated measures was performed and Tukey’s post hoc test (α = 5%) was used. The surface roughness increased in groups G2 and G3 as a function of time, whereas microhardness and mass measurements demonstrated a significant reduction for groups associated with challenges. Ultrastructural evaluation indicated a loss of the aprismatic layer and exposure of the enamel prisms in all groups after 14 days of bleaching, with more pronounced results in G2 and G3 after 28 days. In conclusion, abrasive and erosive challenges potentiated the deleterious effects of tooth whitening on microhardness, ultramorphology, and mass, without affecting the roughness of dental enamel.

## 1. Introduction

Home tooth bleaching consists of daily self-application of a bleaching gel by patients using an individual tray for a predetermined period and must be performed under the supervision of a dentist [1]. Low concentrations of peroxides (10–22% carbamide peroxide or 3.5–10% hydrogen peroxide) are used for a longer application time compared to the office technique [2,3]. However, in some countries, these bleaching products do not require a prescription, thus the patient can easily use them without professional supervision, and may exceed the usage time in an attempt to obtain whiter teeth [4].

Studies [5,6,7] have demonstrated the importance of not exceeding the recommended time for using bleaching gel, as its indiscriminate use allows for the product to act not only on chromatogenic organic substances, but also on the mineral structure of the tooth, causing structural changes in the enamel [5] such as depression, porosity, and transient modifications in the physical properties and chemical composition of the substrate [7,8,9].

Some patients may be dissatisfied with the long period of daily use of 10% carbamide peroxide for home dental bleaching, leading professionals to resort to products with higher concentrations (15–22%) to reduce the daily treatment time [10]. However, the action of peroxides appears to be intricately linked to the concentration of the gel and primarily to the time of exposure. With an increase in these variables, the oxidation process is more intense, possibly resulting in complications and exacerbated effects [4,10,11,12,13]. However, other studies have shown that lower concentrations of bleaching gel do not change the mass measurement [6] or surface roughness (SR) [5], but can affect the enamel microhardness (MH) [5].

Concomitantly with the possibility of prolonged at-home tooth bleaching, which alone can cause damage [9], modern society characteristically presents a significant intake of acidic beverages (the main etiological factor for dental erosion) due to a new lifestyle. In addition, a greater concern for self-care with regard to oral health is observed [4,14], mainly represented by the action of teeth brushing. These common daily habits (consuming acidic beverages and brushing teeth) have been proven to exacerbate the aforementioned adverse effects [13,15]. Ribeiro et al. evaluated the association of daily habits with excessive bleaching with a low-concentration gel and showed that the erosive action was more harmful to dental enamel [15].

Considering the lack of evidence regarding the effects of the excessive exposure of dental enamel to 22% carbamide peroxide gel (high-concentration home bleaching gel) associated with erosive and abrasive challenges, this study aimed to evaluate the influence of these associations on the ultrastructure and surface properties of the dental enamel.

## 2. Materials and Methods

### 2.1. Ethical Aspects and Sample Definition

This study was approved by the Ethics Committee on Animal Use (No. 7237291019). A cooperative where animals are slaughtered for meat consumption donated 110 sound bovine incisor teeth (Bos taurus indicus). The teeth were disinfected by immersion in a 0.1% thymol solution for 1 week, followed by the removal of the periodontal tissue. They were then analyzed under a stereomicroscope, and teeth with cracks or surface defects were discarded.

### 2.2. Teeth Specimens

Dental crowns were sectioned in two places using a double-sided diamond disc. The first and second cuts were placed at a distance of 15 mm and 5 mm, respectively, from the cemento-enamel junction; thus, samples were obtained from the middle portion of the dental crown with a height of 10 mm. The specimens used for the MH and SR tests were embedded in a PVC matrix with self-curing resin, and after 24 h, the buccal surface was flattened using a polisher with cooling to standardize the surface of the dental enamel. In the samples used for the analysis of mass measurement and ultrastructure, all of the remaining dentin was removed, leaving only a thin layer of enamel. Finally, the samples were washed with distilled water in an ultrasonic bath for 2 min. The methodological steps of this study are illustrated in Figure 1.

### 2.3. Tooth Bleaching

The bleaching agent used was 22% carbamide peroxide (Whiteness Perfect 22%; FGM, Joinville, SC, Brazil), applied daily for 1 h, with a ratio of 0.1 mL of bleaching gel to 0.05 mL of artificial saliva [16], dispensed on acetate trays made previously for each specimen to facilitate the gel application and ensure intimate contact of the gel with the tooth structure. The bleaching protocol recommended by the manufacturer was 14 days, and twice the time (28 days) was considered excessive, according to the literature [9]. During the bleaching period (1 h), specimens were stored in a biological oven at 37 °C to simulate the temperature of the oral environment. Thereafter, the specimens were washed with an air/distilled water jet for 1 min, and placed approximately 5 cm from the enamel surface exposed to the intervention. Then, the specimens of the groups subjected to tooth whitening alone were immersed in artificial saliva and placed in a biological oven (37 °C/24 h), while the others were subjected to subsequent treatments (erosive and abrasive challenges). Table 1 lists the compositions of the materials used.

### 2.4. Erosive Challenge

Approximately 200 mL of Gatorade^®^ (PepsiCo Inc., Purchase, NY, USA), an isotonic drink, was kept in sterile Becker flasks. After bleaching, specimens were immersed in the solution for 10 min under light manual agitation (total of two 10 s shakes, with a 5 min interval between them, performed by the same operator) once a day for 28 days. After each immersion cycle, specimens were washed with distilled water and stored in artificial saliva (37 °C) in a biological oven. The pH of the drink was measured (pH = 3) before and after at each immersion/agitation cycle using a portable pH meter. The temperature of the liquid was measured using the same equipment used to check the standard temperature, which was maintained at 25 °C. The drink was discarded at the end of each daily cycle [17].

### 2.5. Abrasive Challenge

At the end of the 14th and 28th day of treatment, the specimens were exposed to abrasive challenges. An Oral B Professional Care 500 electric toothbrush (Oral B, Schwalbacham Taunus, Germany), fixed on an appropriate support, was used in “continuous mode”. On both days, each enamel surface received a single cycle of 210 s (to simulate three daily brushings on each tooth, 15 s daily), with a load of 2 N on the brushes [18]. During brushing, a solution of 3 g toothpaste and 0.3 mL artificial saliva [18] was prepared at the exact moment of the test, without modification of the active principles, and it was deposited on each specimen. At the end of brushing, the specimens were removed, washed for 30 s with distilled water, immersed in artificial saliva, and stored in a biological oven (37 °C/24 h).

### 2.6. Microhardness and Roughness Tests

Dental fragments for SR and MH evaluation were individually embedded in chemically activated acrylic resin using 11 mm-high acrylic matrices, leaving their buccal surface exposed to the external environment. After 24 h, the vestibular surfaces of the specimens were flattened using a horizontal semi-automatic polisher, with sandpapers in decreasing granulation order: #400 for 30 s, #600 for 30 s, #1500 for 1 min, and #2000 for 1 min; prior to each sandpaper change, the specimens were washed for 3 min in an ultrasonic bath to avoid interference from the previously used sandpaper.

SR and MH analyses were performed on different areas of the same specimen (*n* = 15). The enamel surface of the specimen was divided into four parts: one for the evaluation of the SR and three for the microhardness analyses; thus, for the latter, there was no coincidence of the already indented areas. The readings were performed as follows: T0, before the beginning of the bleaching protocol (negative control); T1, at the end of the 14th day of bleaching (positive control); T2, at the end of the 28th day of bleaching.

SR was evaluated using a rugosimeter (SJ-301; Mitutoyo, Los Angeles, CA, USA). The adopted parameter for calculating SR was the arithmetic roughness determined by the average of three readings, with a tracing limit of 5 mm and a sample length or cut-off of 0.25 mm.

Knoop MH was measured using a microdurometer (HMV-2; Shimadzu, Kyoto, Japan) with three indentations spaced 100 µm apart and a load of 50 g for 20 s. For each group, readings were taken at T0, T1, and T2. For each specimen, the arithmetic mean of three readings was recorded as the MH value.

### 2.7. Mass Variation Evaluation

To analyze mass variation (*n* = 10), teeth were additionally sectioned in the mesiodistal direction to separate the buccal and lingual portions, and the latter was excluded. The vestibular dentin layer was removed using a diamond bur with a high-speed handpiece under constant irrigation, leaving only the vestibular dental enamel. Then, all fragments were washed in an ultrasonic bath with distilled water for 20 min, embedded in matrices with proper molding to receive the treatments, and undocked to determine the weight.

The weights of the fragments were determined using an electronic analytical balance with an accuracy of 0.0001 g at T0, T1, and T2. In each group, the specimens were washed after treatment using an air/distilled water spray (1 min), from a distance of approximately 5 cm from the enamel surface, and dried using a Philco Titanium Travel dryer for 2 min, the time necessary for there to be no further change in the weight of the specimens [6], and the final weight was determined.

### 2.8. Ultrastructure Analysis

At each analysis time (T0 as a negative control group, *n* = 5; T1 and T2, *n* = 5 for each group), scanning electron microscopy (SEM) of the central area of the specimens was conducted. The metallization comprised of the vacuum precipitation of a micrometric film of a platinum alloy on the surface of the dental enamel. Electron micrographs were obtained using a TESCAN electron microscope, model Mira3 (Tescan Ltd., Brno, Czech Republic), with a field-emission gun-type electron gun. It was possible to perform a morphological analysis of the enamel surface at different times of treatment through visual and qualitative examination of the images.

### 2.9. Statistical Analysis

Statistical calculations were performed using the GraphPad Prism 7.0 software (GraphPad Software Inc., La Jolla, CA, USA). After verifying normality, analysis of variance of two factors was used for repeated measures, with the post hoc Tukey test for SR, MH, and mass variation. The α level of significance (*p* ≤ 0.05) was used for all analyses.

## 3. Results

### 3.1. The Roughness in the Enamel Surface Was Modified Only in the Erosive and Abrasive Challenge

In the excessive bleaching + erosive challenge group (G2) and excessive bleaching + erosive challenge + abrasive challenge group (G3), the SR increased at T1 and T2 compared to that at T0, (*p* < 0.05), with the highest mean values in G3. In the excessive bleaching group (G1), values at T0, T1, and T2 were similar. Figure 2 shows the average SR.

### 3.2. The Microhardness Decreased at All Times in the Erosive and Abrasive Challenges, with Lower Values at 28 Days

The results demonstrated that there was a statistically significant difference between all treatment times for G2 and G3 (*p* < 0.0001), while in G1, the greatest loss was observed in the comparison between T0 and T2 (*p* < 0.05). Figure 3 shows the MH values.

### 3.3. Bleaching Alone Decreased the Mass in an Excessive Regimen, While Erosive and Abrasive Challenges Affected the Enamel after 14 Days of Treatment

There was a significant time-dependent reduction in the enamel mass (*p* < 0.05) in groups G2 and G3. Bleaching alone (G1) affected the enamel in the excessive regime (T2). Figure 4 illustrates the mean (standard deviation) mass variation.

### 3.4. Ultrastructural Damage Was More Evident in the Excessive Bleaching Combined with Erosive and Abrasive Challenges Groups

At T0, specimens did not receive any treatment; therefore, it was possible to observe the integrity of the enamel prisms. At T1, some degree of exposure of the enamel prisms, characterizing the loss of aprismatic enamel in the horizontal direction, was observed, which was more evident in G2 and G3. Figure 5 shows the electron micrographs of all treatments.

## 4. Discussion

This study demonstrated in an unprecedented manner that tooth bleaching with 22% carbamide peroxide for up to 28 days, combined with an isotonic acidic drink and tooth brushing, significantly affected the enamel MH, SR, mass, and ultrastructure, whereas isolated bleaching therapy did not affect the enamel SR and induced less pronounced ultrastructural changes.

Peroxides used for an extended period can cause the enamel matrix to dissolve, change the chemical composition [9], and lead to the loss of enamel mass [6]. In the present study, bleaching was performed using 22% carbamide peroxide gel for up to 28 days. A reduction in MH and mass was observed in all groups tested due to the increase in bleaching time. In G1, SR was not altered by bleaching, suggesting that saturation of the artificial saliva and the remineralizing component of the gel may have led to the adsorption of calcium and phosphate on the tooth surface [9]. Additionally, the neutral pH of the gel may have influenced the results. These results are similar to those of Vilhena et al. [9], who demonstrated that home bleaching using 10% carbamide peroxide for a prolonged time did not affect the SR. In that study, the bleaching gel used had sodium fluoride in its composition, in addition to the artificial saliva that had been used as a storage environment.

In contrast, in groups G2 and G3, the SEM images showed that the erosive challenge compromised the interprismatic enamel, which was more susceptible to demineralization [14]. This suggested that the aforementioned remineralizing components could not reverse the changes on the enamel surface during the abrasion and erosion processes.

In this study, the groups that underwent bleaching and were exposed to an isotonic drink (Gatorade^®^) had the most pronounced results with a reduction in the MH and mass (*p* < 0.05), and an increase in SR (*p* < 0.0001). This can be explained by the fact that the isotonic drink (Gatorade^®^) has an acidic pH (≈3.0), and its erosive potential is well-documented in the literature [19,20]. Furthermore, in addition to acid challenges, our teeth undergo daily mechanical brushing, which is considered as a safe and effective therapy for maintaining oral health [21]. However, in the present study, tooth brushing appeared to increase the SR caused by the acid challenge, which is an undesired outcome, as rough surfaces favor the greater deposition of biofilm. Previous studies have shown that the threshold for bacterial retention is 0.2 μm [22].

In general, toothpastes contain sodium fluoride, which is the primary remineralizing agent used for caries prevention. It also assists in controlling dental erosion because of its ability to reduce the solubility of the surface and assist in mineral recovery [20]. Studies have demonstrated that manual brushing alone does not cause significant damage to the topography and mechanical characteristics of the enamel; however, brushing may damage the enamel when performed soon after erosive challenges [21]. In G3, where brushing was combined with the erosive challenge, the outermost layer of enamel appeared to be degraded by the acid action, and a more consistent enamel appeared to replace it, thus contributing to higher MH compared to that in G2.

Clinically, the MH and SR results obtained in G3 may be relevant, and it is up to the dentist to guide patients who are undergoing bleaching procedures regarding the consumption of acidic foods and drinks since feeding and tooth brushing are part of their daily activities.

A study evaluating the susceptibility of demineralized enamel to abrasion demonstrated that electric toothbrushes can lead to a significantly greater loss of surface compared to manual brushing [23]. Therefore, we believe that in addition to the erosion process, the exposure of the interprismatic spaces (visualized by SEM) of the eroded enamel in G3 may be associated with this fact.

However, a 2019 study [6] evaluating the variation in dental enamel mass with prolonged use of 10% carbamide peroxide did not find any loss of mass, which is in contrast to the findings of the present study. This can be explained by the higher concentration of bleaching gel used in this study, which may have caused more pronounced damage, as illustrated by the SEM images.

## 5. Conclusions

Within the limitations of this study, it is possible to appreciate that:
1.Supervised at-home tooth bleaching with a high-concentration (22% carbamide peroxide) gel did not affect the SR, even when performed excessively. Tooth bleaching with a high-concentration (22% carbamide peroxide) gel negatively affected the MH and mass of the dental enamel only when performed excessively (28 days).2.Tooth bleaching combined with exposure to an acidic beverage significantly increased the SR, with a loss of enamel mass and MH.3.Excessive tooth bleaching combined with exposure to acidic drinks and tooth brushing exacerbated the deleterious effects on the enamel, qualitatively evidenced by the visual inspection of the enamel ultrastructure.

## Figures and Tables

**Figure 1 materials-15-07496-f001:**
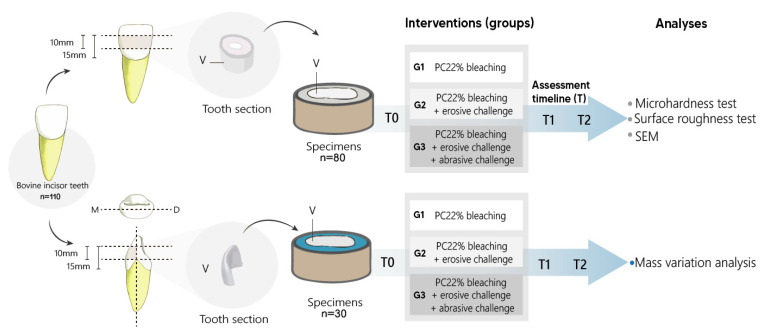
The preparation of specimens and group division.

**Figure 2 materials-15-07496-f002:**
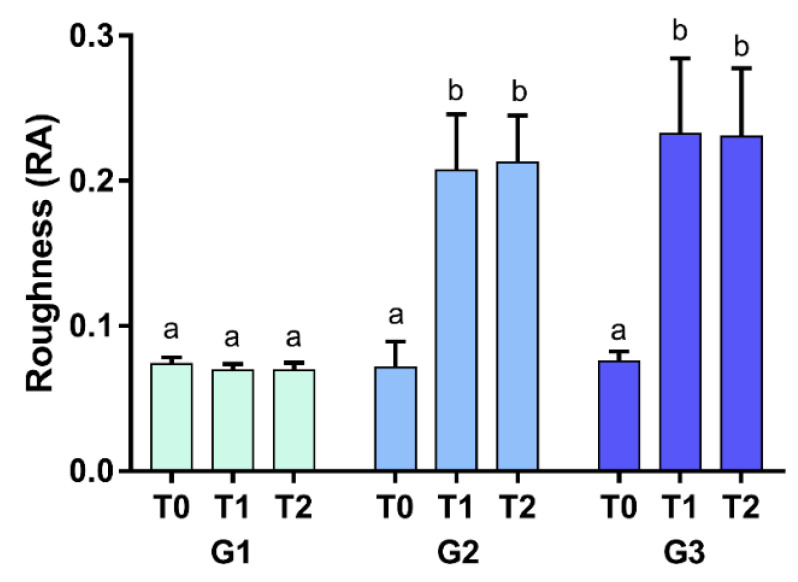
The mean ± standard deviations of the surface roughness submitted to two-way analysis of variance for repeated measurements according to the periods for the groups tested. Different lowercase letters indicate an intragroup statistically significant difference.

**Figure 3 materials-15-07496-f003:**
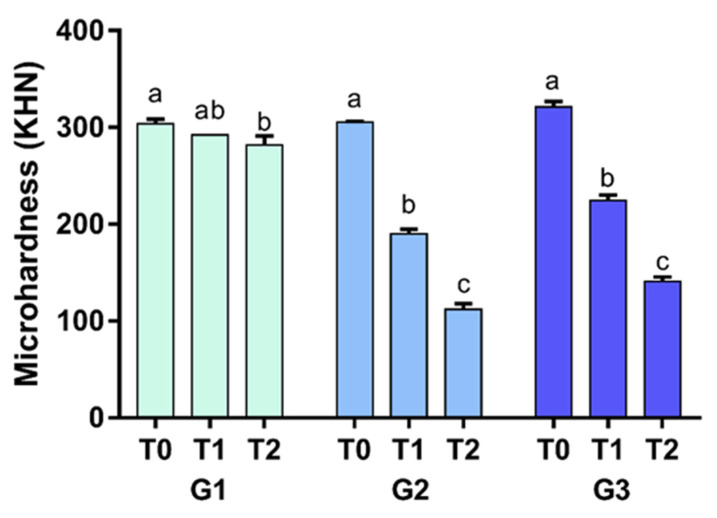
The mean ± standard deviation of the microhardness subjected to two-way analysis of variance for repeated measurements according to the periods for the groups tested. Different lowercase letters indicate an intragroup statistically significant difference.

**Figure 4 materials-15-07496-f004:**
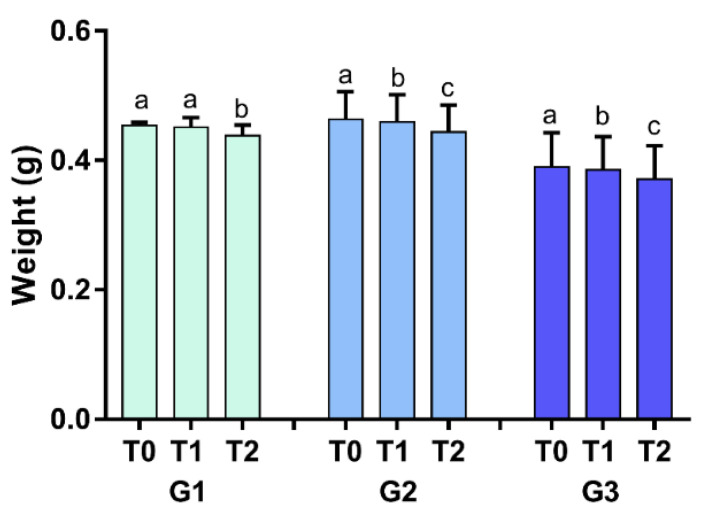
The mean ± standard deviation of the mass variation evaluated by two-way analysis of variance for repeated measures according to the periods for the groups tested. Different lowercase letters indicate an intragroup statistically significant difference.

**Figure 5 materials-15-07496-f005:**
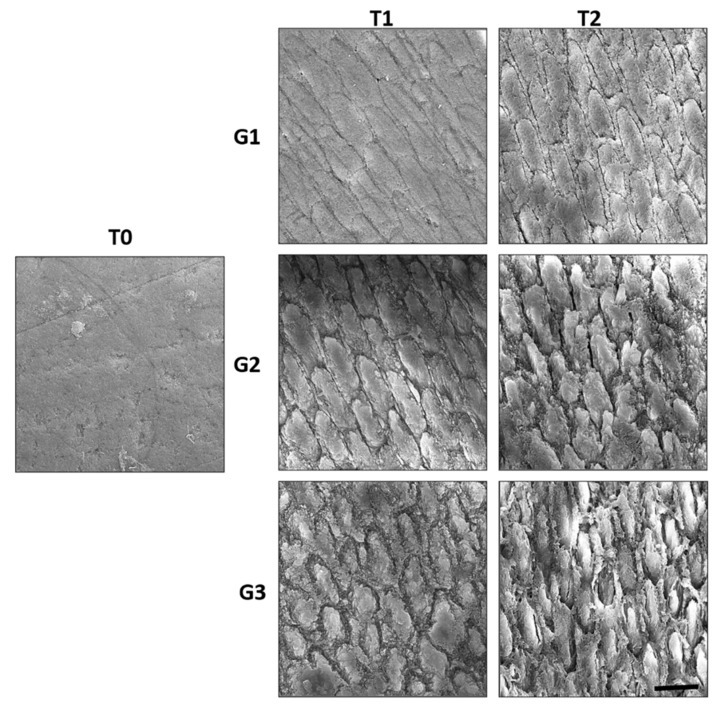
The scanning electron microscopy images representative of the effect of the following treatments with time on dental enamel: T0 without treatment (artificial saliva), G1-T1 CP22 for 14 days, G2-T1 CP22 + Gatorade for 14 days, G3-T1 CP22 + Gatorade + simulated toothbrushing for 14 days. T2 images following the same treatments for 28 days. Scale Bar: 5 µm. CP22, 22% carbamide peroxide.

**Table 1 materials-15-07496-t001:** The description of materials used including their trade names and manufacturers.

Material (Manufacturer)	Composition
Whiteness Perfect bleaching gel (FGM Produtos Odontológicos Ltd.a, Joinville, SC, Brazil)	22% carbamide peroxide, carbopol, potassium hydroxide, sodium fluoride, glycerol, deionized water, and pH around 7.
Artificial saliva (A fórmula—Farmácia de Manipulação, Belém, PA, Brazil)	Sodium bicarbonate 2190 mg, potassium phosphate 1270 mg, magnesium chloride 125 mg, calcium chloride 441 mg, potassium chloride 820 mg, sodium fluoride 4.5 mg, nipazol 100 mg, carboxymethylcellulose 8 mg, distilled water 3000 mL.
Colgate toothpaste Total 12 (Colgate-Palmolive, São Bernardo do Campo, SP, Brazil)	Sodium fluoride (1450 F ppm), water, sorbitol, hydrated silica, sodium laurylsulfate, PVM/MA copolymer, flavor, carragenine, Sodium hydroxide, triclosan, titanium dioxide (Cl 77,891), dipenteno and RDA 70.
Gatorade^®^ isotonic of citrus fruits (PepsiCo Inc., Purchase, NY, USA)	Water, sucrose, glucose, sodium chloride, sodium citrate, monobasic potassium phosphate, acidulant (citric acid), natural orange and grapefruit aroma, artificial colors, and pH = 3.

## Data Availability

Not applicable.

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
