# Peer review of "Excessive Dental Bleaching with 22% Carbamide Peroxide Combined with Erosive and Abrasive Challenges: New Insights into the Morphology and Surface Properties of Enamel"

_materials, 2022, doi:10.3390/ma15217496_

Round 1

Reviewer 1 Report

The study addresses the excessive dental bleaching with 22% carbamide peroxide as-2 associated with erosive and abrasive challenges: new insights 3 in morphology and surface properties of enamel. The authors used experimental and optimization methods to show the results.

Below are my constructive criticisms of the article.

  1. Abstract should start with the word “The abstract should start with the word, it is difficult to understand whether authors talking about their work or existing work.
  2. Avoid brackets information in the abstract.
  3. Why the authors considered G and T as the notations, please justify. I recommend adding notations and symbols.
  4. I do not find challenging literature in the introduction and also there is a only two or three papers from last three years. I recommend increasing the literature with the latest work and exploring a research gap in a better a way.
  5. In the methodology, the authors divided into nine subsections, and I believe each section is a type of analysis. Therefore, before proceeding subsections please a smart char to overview the analysis and its importance to current work.
  6. Where is the methodology of optimization method such as ANOVA and others? Please add in the methodology and illustrate why it is important to current work.
  7. Did you follow any standard parameters and its variation? What are the levels and parameters, please add to the methodology section?
  8. I did not find result section has any novelty. Simply charts have been shown with SEM.
  9. Discussion, it needs patience to read and connect to the results. I suggest merging results and discussion.
  10. The conclusion is poorly written, please update it.

Author Response

    We are very grateful for the great contribution that your suggestions have added to the article. Follow the changes and comments.

  1. Abstract should start with the word “The abstract should start with the word, it is difficult to understand whether authors talking about their work or existing work.

Considering the reviewer’s suggestion, the first sentence of the abstract was adjusted as: “It was evaluated the effects…”

2. Avoid brackets information in the abstract.

We appreciate the suggestion. The square brackets have been removed from the summary.

3. Why the authors considered G and T as the notations, please justify. I recommend adding notations and symbols.

We used G to describe the various groups and T to describe the 3 times in which the readings were carried out (letters used as initials), authors aimed to make the text more didactic and shorter.

4. I do not find challenging literature in the introduction and also there is a only two or three papers from last three years. I recommend increasing the literature with the latest work and exploring a research gap in a better a way.

Thanks for the contribution. More recent references have been included and the main purpose has been better placed in the introduction in paragraph 4.

5. In the methodology, the authors divided into nine subsections, and I believe each section is a type of analysis. Therefore, before proceeding subsections please a smart char to overview the analysis and its importance to current work.

The authors thank to reviewer’s observation. However, it is important to point out that each subsection is not, necessarily, an analysis. Therefore, the division of methodology’s section into 9 subsections was performed to facilitate the comprehension of the methods used, as well as aspects related to ethical and statistical consideration, which are not present in figure 1.

6. Where is the methodology of optimization method such as ANOVA and others? Please add in the methodology and illustrate why it is important to current work.

The two-way ANOVA statistical test was used to show which factor (brushing or erosion) most influenced the enamel properties, measured by the roughness, hardness and mass measurement tests.

7. Did you follow any standard parameters and its variation? What are the levels and parameters, please add to the methodology section?

The 95% confidence interval, with a 5% significance level.

These parameters are described in subsection 2.9 Statistical analysis.

8. I did not find result section has any novelty. Simply charts have been shown with SEM.

The authors thank to reviewer’s observation. However, it is important to point out that results’ section was divided into 4 subsections, considering the main result observed for each methodological test used (roughness, microhardness, enamel mass variation and ultrastructural analysis), emphasizing the scientific contribution achieved, what were better explained in discuss’ section (see the example below).

Considerations for SEM are mentioned in the discussion section, ex.:

“In contrast, it is possible that for groups G2 and G3, these remineralizing components were unable to overcome the changes on the enamel surface during the abrasion and erosion processes, which was evident in the images obtained using SEM, where the erosive challenge compromised interprismatic enamel, which was more susceptible to demineralization [17]”

9. Discussion, it needs patience to read and connect to the results. I suggest merging results and discussion.

We agree with the reviewer, that merging the results with the discussion optimizes space and information. However, the scope of the current journal as well as journals suggest using separate results for discussion.

10. The conclusion is poorly written, please update it.

Following the guidance, the conclusion was rewritten

Dental bleaching with CP22 used in an excessive regimen was able to reduce enamel’s microhardness and mass, altering substrate’s ultramorphology as well. Moreover, erosive and abrasive challenges contributed to potentiate these changes. 

Reviewer 2 Report

Review report on the topic ‘Excessive Dental Bleaching with 22% Carbamide Peroxide Associated with Erosive and Abrasive Challenges: New insights in Morphology and Surface Properties of Enamel’. Comments are listed below:

  1. Strengthen the abstract section. Add the key conclusion of the works in the last two lines of the abstract section.
  2. Discuss the motive behind the work. The clear application of the work should be discussed in the introduction section.
  3. There are numerous spelling and grammatical errors. Please revise the manuscript thoroughly.
  4. The novelty of the work is not clear.
  5. Try to make a bridge between current and previously published work and specify the gap area and objective of the work.
  6. Experimental information is provided in a rough manner. Add detailed things and provide good quality image.
  7. Section 2.4 and 2.5 are inserted forcefully. No need to discuss the unnecessary information in the experimental section.
  8. Hardness varies, that part is ok but discussion behind the hardness variation is not discussed.
  9. The quality of image 5 is very poor also, the scale is missing. Provide good qualitative and quantitave analysis. Add EDS spectra area map and line map along the boundaries and inside the matrix.
  10. Add key bullet points for both future trends and conclusions.

Author Response

Thank you very much for the suggestions that contributed greatly to the improvement of the article.

  1. Strengthen the abstract section. Add the key conclusion of the works in the last two lines of the abstract section.

The summary conclusion has been changed to:

“Abrasive and erosive challenges potentiated the deleterious effects of tooth bleaching on micro-hardness, ultramorphology and mass measurement, however without affecting the roughness of dental enamel.”

2. There are numerous spelling and grammatical errors. Please revise the manuscript thoroughly.

The paper initially went through a company specializing in language proofing and was proofread again now by a native English speaker.

3. Discuss the motive behind the work. The clear application of the work should be discussed in the introduction section.

“Concomitantly with this possibility of prolonged home dental bleaching, modern society characteristically has a significant intake of acidic drinks (main etiological factor for dental erosion), in addition to a higher concern of self-care in oral health [4], represented primarily by the action of tooth brushing; these (ingestion of acidic drinks and toothbrushing) common daily habits can exacerbate the adverse effects aforementioned [13]”

4. The novelty of the work is not clear.

We appreciate your observation. In this sense, the literature has not evaluated excessively bleached enamel, which is exposed to an isotonic of worldwide consumption associated with daily toothbrushing.

5. Try to make a bridge between current and previously published work and specify the gap area and objective of the work.

Dear reviewer, we appreciate your post.

Previous work (Vilhena et al., 2019), evaluated only the effects of excessive whitening of tooth enamel. Excessive tooth bleaching was shown to alter micromorphology and hardness but without altering enamel roughness. The present study evaluates, also excessively, tooth whitening with a higher concentration gel, exposing this same enamel to isotonic and toothbrushing.

6. Experimental information is provided in a rough manner. Add detailed things and provide good quality image.

We appreciate the observation and some information that is underlined in yellow has been added for better clarification. For example:

“…the specimens that were destined for the microhardness and roughness tests were embedded in a PVC matrix with self-curing resin, and after 24 hours the buccal surface was flattened in polisher with cooling to standardize the surface of the dental enamel. The samples that were destined for the analysis of mass measurement and ultrastructure had additional wear of all the remaining dentin, leaving only the thin layer of enamel.”

Regarding the methodological figure, it was uploaded in better format and quality as a supplementary file.

7. Section 2.4 and 2.5 are inserted forcefully. No need to discuss the unnecessary information in the experimental section.

We understand your observation. However, the erosive (isotonic drink) and abrasive challenges are not contained in the methodological figure and we believe that for a good understanding of the step-by-step reproduction of the methodology, these steps should be described.

8. Hardness varies, that part is ok but discussion behind the hardness variation is not discussed.

For the results topic, we have placed subsections with the main results of each analysis to draw the reader's attention. In the text of the current study, we split the results of the discussion. The results are better explained in the discussion section, as in the passages below:

“In this study, we found that the groups that underwent the bleaching process and were exposed to the isotonic drink (Gatorade®) had the most expressive results with the reduction of microhardness and mass (p <0.05), as well as an increase in roughness (p <0.0001). This can be explained by the fact that the isotonic drink (Gatorade®) has an acidic pH (≈3.0), and its erosive potential is well documented in the literature [18,19]”

“In G3, where brushing was associated with the erosive challenge, the outermost layer of enamel appeared to be degraded by the acid action, and a more consistent enamel appeared to replace it; thus, contributing to higher values of microhardness when compared to the average of G2.”

9. The quality of image 5 is very poor also, the scale is missing. Provide good qualitative and quantitave analysis. Add EDS spectra area map and line map along the boundaries and inside the matrix.

We appreciate the observation and have already corrected figure 5 by inserting the scale in the legend. the evaluation by the scanning electron microscopy method in the present study had the intention of a qualitative analysis only. In this sense, a quantitative analysis such as the EDS was not performed.

10. Add key bullet points for both future trends and conclusions.

We believe that the main contribution of this study is for dentists to keep teeth whitening under their supervision and to warn patients who have very acidic diets to take greater care during the bleaching treatment, respecting the time for brushing after drinking acidic.

Reviewer 3 Report

The article is focused on effect of excessive dental bleaching with 22% of carbamide peroxide and evaluates changes in upon erosive and abrasive impacts.

regarding the Abstract

Conclusions in the abstract shall be written more clear.

Regarding the Introduction chapter

First 13 references are correct however I recommend to reference some more newer publications from the recent three years.

Regarding the chapter Materials and methods

the figure #1 Shelby rented so the text could be larger and readable this figure needs to be redrawn.

Table number 1 does not have proper design as defined in the template for the articles to journal Materials.

in the line 95 there's a registered trademark “Gatorade®” mentioned it needs to be properly referenced

regarding the discussion chapter

This chapter is well written I have no special remarks

regarding the conclusions

I do recommend fragmenting the long sentence online 282 283 and 284 to make it more readable.

In general the manuscript is well written and describes interesting and useful topic with minor changes I recommend it for publication.

Author Response

regarding the Abstract

Conclusions in the abstract shall be written more clear.

We appreciate the observation and the conclusion has been modified in order to meet the objectives and be clearer to the reader.

“In conclusion, the abrasive and erosive challenges potentiated the deleterious effects of tooth bleaching on microhardness, ultramorphology and mass measurement, however without affecting the roughness of dental enamel.

Regarding the Introduction chapter

First 13 references are correct however I recommend to reference some more newer publications from the recent three years.

 Thanks for the note, and more recent articles have been included.

Regarding the chapter Materials and methods

The figure #1 Shel by rented so the text could be larger and readable this figure needs to be redrawn. 

Thank you for your observation and we appreciated your suggestion for our study. Thus, Fig.1 was restructured and now we hope that has been clear

Table number 1 does not have proper design as defined in the template for the articles to journal Materials.

 Thanks for the observation. Table 1 adjusted.

In the line 95 there's a registered trademark “Gatorade®” mentioned it needs to be properly referenced

 Reference placed; product is also described in table 1.

Regarding the discussion chapter

This chapter is well written I have no special remarks

regarding the conclusions

I do recommend fragmenting the long sentence online 282 283 and 284 to make it more readable.

 Conclusion section as suggested, has been rewritten.

“Dental bleaching with CP22 used in an excessive regimen was able to reduce enamel’s microhardness and mass, altering substrate’s ultramorphology as well. Moreover, erosive and abrasive challenges contributed to potentiate these changes.”

Round 2

Reviewer 1 Report

Thanks for authors for amending the suggestions but still there are two key areas not clearly covered. 

1. Please add the historic perspective of the topic. Support it from previous research and a table presenting it would be a good contribution. 

2. The conclusion can be extended. It is very straightforward at the moment. 

Author Response

We appreciate your collaboration:

  1. Please add the historic perspective of the topic. Support it from previous research and a table presenting it would be a good contribution.

      The suggestion was inserted in lines 49 to 51 and 58 to 60, giving a historical explanation of the research line.

2. The conclusion can be extended. It is very straightforward at the moment. 

Modified conclusion:

"Within the limitations of this study, it is possible to appreciate that:

  1. Supervised at-home tooth bleaching with a high-concentration (22% carbamide peroxide) gel did not affect SR, even when performed excessively. Tooth bleaching with a high-concentration (22% carbamide peroxide) gel negatively affected the MH and mass of dental enamel only when performed excessively (28 days).
  2. Tooth bleaching combined with exposure to an acidic beverage significantly increased SR, with a loss of enamel mass and MH.
  3. Excessive tooth bleaching combined with exposure to acidic drinks and tooth brushing exacerbated the deleterious effects on enamel, qualitatively evidenced by the visual inspection of the enamel ultrastructure"

Reviewer 2 Report

Accepted.

Author Response

We thank you for your great contribution to our manuscript and a new language review was carried out (attached certificate)
